# Research on Cooperative Behavior of Green Technology Innovation in Construction Enterprises Based on Evolutionary Game

**Qing'e Wang** [1] , **Wei Lai** [2,*] , **Mengmeng Ding** [1] and **Qi Qiu** [1]

1. School of Civil Engineering, Central South University, Changsha 410075, China; wqecsu@csu.edu.cn (Q.W.); dmmcsu@csu.edu.cn (M.D.); 174801042@csu.edu.cn (Q.Q.)
2. The Second Xiangya Hospital of Central South University, Changsha 410011, China
*  Correspondence: laiweixy7919@csu.edu.cn

**Abstract:** The dynamic evolution game model is built by using evolutionary game theory, and the evolutionarily stable strategy is analyzed by matlab2018b software in this paper. The cooperation willingness, sharing level, income distribution, and punishment mechanism are comprehensively considered in this model, and numerical simulations of the influence of various influencing factors on the cooperation strategy selection of green technology innovation for construction enterprises are carried out. Then, countermeasures and suggestions are put forward. The results of evolutionary game analysis show that the cooperation willingness, sharing level, income distribution, and punishment mechanism have a significant impact on the cooperative evolution direction of green technology innovation for construction enterprises, separately. Stronger cooperation willingness or higher relative value of positive spillover, or reasonable income distribution can promote partners to adopt active cooperative strategies, while appropriately increasing punishment intensity can prevent opportunistic behaviors and improve the probability of success of cooperative innovation.

**Keywords:** green technology innovation cooperation; evolutionary game theory; numerical simulation

## 1. Introduction

There are still defects in traditional building materials, techniques and design concepts. The rapid expansion of China's construction industry has led to a large amount of energy consumption [1,2]. Construction waste and carbon emissions continue to be produced, and the total carbon emissions account for up to 51.3% of national carbon emissions. In recent years, driven by national policies and market demand, green development has become an important direction for the development of the construction industry [3], and green technology innovation has become a key driving force for the transformation and development of construction enterprises.

Brawn and Wield collectively referred to technologies, processes, and products that can reduce the consumption of raw materials and energy and mitigate environmental pollution as green technologies in 1994 [4]. The green technology innovation achievements commonly include energy, water conservation, material saving, and recycling [5]. Combined with relevant policies of government departments, the current green technology innovation activities in China are characterized by external influence, gradual development, and cross-border integration [6]. Different from the traditional technological innovation with economic benefits as the main pursuit goal, green technological innovation focuses more on the unity of economic, social, and ecological benefits [7], which is the key for construction enterprises to realize their green development. However, green technology innovation activities usually require long-term investment of personnel and funds, as well as the need to take great risks [8,9], and there is a strong economic externality [10]. Therefore,

the development of green technology innovation has certain obstacles. Most construction enterprises in China choose to give up green technology innovation, which leads to the lagging development of green technology innovation in China's construction industry compared with that in developed countries. To solve the above dilemmas, construction enterprises usually carry out green technology innovation through cooperation.

In an earlier study on the connotation of innovation cooperation, Fusfeld et al. (1985) argued that innovation cooperation refers to a "cooperative contract" formed by two or more collaborators with the same R&D goals and complementary resources, which can effectively promote the achievement of innovation results [11]. Fu Jiaji (1998) further enriched the definition of innovation cooperation and broadened the scope of innovation cooperation to include enterprises, research institutions, schools, etc. The connotation of innovation cooperation specifically covers: based on the premise that the pursued interests and goals are the same, resources are shared and complementary, the parties cooperate in the whole process or part of the process of technological innovation according to the pre-agreed rules of cooperation. The costs, risks, and results of technological innovation are shared [12]. In recent years, with the support of policies and the improvement of enterprises' innovation capabilities, the willingness of cooperation among construction enterprises is increasing [13,14]. However, in the process of cross-organizational cooperation, there are various risks, which lead to the opportunistic behavior of construction enterprises, resulting in the rupture of cooperation relations and the failure to achieve successful innovation results [15].

At present, scholars have conducted extensive research on enterprise green technology innovation and innovation cooperation, but mostly in the manufacturing and energy industries [16]. In the green technology innovation research literature, the influencing factors, barriers and evaluation have become major hot spots in recent years [17,18]. In the research of enterprise innovation cooperation, scholars have increasingly studied the influencing factors, risks and strategy choices [19]. After summarizing the research literature, it was found that there are still some unsolved problems in the existing research as followings:

1. There is a lack of cooperation research related to green technology innovation in the construction field. There have been many studies on innovation cooperation, and the number of research objects is gradually increasing, but there are fewer studies in the construction field. At present, most of literatures related to the green technology innovation of construction enterprises remain in the aspects of the principle analysis and practical application of a certain green building technology [20]. However, with the concern for environmental problems such as energy loss and pollution damage in the construction industry, its cooperative research has important practical significance to promote the application and development of its green technology innovation.

2. There is a lack of research on the construction enterprise as the main body of cooperation. Since construction enterprises often do not occupy a dominant position in the innovation process, the existing research is mostly conducted for other cooperation subject types, such as industry–university research innovation cooperation and university research cooperation alliances [21]. However, in recent years, with the trend of promoting enterprises as the main body of innovation, it is necessary to study inter-enterprise innovation cooperation.

3. The research on the influencing factors of cooperation strategy selection and its mechanism of action is not comprehensive and in-depth enough [22], and the relevant research on construction enterprises is also lacking. Numerous relevant studies of enterprise have found that the innovation capability, experience and reputation of enterprises can affect the cooperative behavior strategy and results through the willingness to cooperate [23]. Secondly, information sharing is one of the important basis for inter-enterprise cooperation and coordination [24]. Adequate sharing among enterprises can produce double spillover effects in different situations [25], which has a positive impact on the choice of cooperative behavior strategies [26,27]. In

addition, academics generally agree that the benefit distribution mechanism is also a key factor affecting cooperative relationships [28]. In the green technology innovation cooperation situation of different types of subjects, such as enterprises, universities and research institutions, benefits and costs also have an impact on cooperation behavior [29]. Different benefit allocation strategies have different effects on the promotion of enterprise cooperation behavior. Therefore, it is important to study how to find the best benefit allocation strategy [30]. The government, as an important intermediary of enterprise innovation cooperation, can promote the efficiency and success of enterprise technology innovation cooperation by adopting reasonable punishment mechanisms [31–33]. Research has confirmed that appropriate government incentives and punishments can play a positive role in promoting cooperation stability [33].

In order to effectively curb the opportunistic behavior in the process of cooperation, it is necessary to comprehensively study the influencing factor of cooperation and its role law on the choice of cooperation strategy of construction enterprises. Therefore, considering the influence of cooperation willingness, sharing level, benefit distribution, punishment mechanism, and other influencing factors on green technology innovation cooperation of construction enterprises, this paper explores the evolutionary law of green technology innovation cooperation behavior of construction enterprises by analyzing the evolutionarily stable strategy.

The paper is organized as follows. In the second part, the basic hypothesis is proposed and the game model is established. In the third part, the main work is to solve the model and analyze the cooperative behavior of construction enterprises in green technology innovation under different conditions. The fourth section covers the numerical simulation model and discussion to verify the accuracy of the evolutionary game model. The last section is the conclusion.

## 2. Model Building

### 2.1. Problem Description and Model Assumptions

In addition to the general enterprise innovation cooperation characteristics, the green technology innovation cooperation among construction enterprises in China has some special characteristics caused by the background of the construction industry as follows.

1. The cooperation object is usually not completely symmetrical. Green technology innovation in the construction industry is usually tailored to the needs of specific engineering projects. The different technical requirements and site conditions of engineering construction can lead to more complex green technology innovation in construction enterprises. In order to improve technology and speed up innovation, the organization involved in green technology innovation cooperation usually comes from a wide range of sources and may differ in terms of enterprise nature, organization type and business operations. This can result in differences in resources, capabilities, knowledge, and technology, which makes it difficult for participants in green technology innovation cooperation to meet the conditions of complete symmetry.

2. Having reciprocity preferences—both the direct goodwill behavior of each other in reciprocity theory (direct reciprocity) [34,35] and the corporate reputation effect formed through a third party (indirect reciprocity) [36] can be used to explain the establishment of trust foundation, thus promoting the green technology innovation cooperation behavior in construction enterprises. Due to the existence of direct reciprocity, successful experiences in engineering project cooperation are beneficial to strengthen the degree of trust among construction enterprises in green technology innovation cooperation. In other words, when enterprises have had successful experiences in cooperation and have established a deeper understanding of each other during the cooperation process, this cooperation history can strengthen trust among enterprises, and enterprises will reciprocate for each other's friendly behavior, prompting the generation of the green technology innovation behavior. In addition, through indirect reciprocity, enterprises can obtain a good corporate reputation and indirect

returns from other beneficiary enterprises [37,38]. In addition, they can establish a strong cooperative relationship with companies that have not had previous contact or cooperation at a lower cost. If the corporate reputation is unknown or unreliable, the enterprises will build trust in the cooperation process by increasing the transaction cost, etc., in order to achieve the purpose of controlling the cooperation risk.

3.  Having a strong spillover effect—green technology innovation cooperation among construction enterprises can generally be carried out in the forms of cooperative engineering projects, technological innovation cooperation results, a joint publication of papers or writing patents, etc. However, since cooperation in engineering projects is the most common cooperation mode among construction enterprises at present, the development of green technology innovation cooperation among construction enterprises is mainly based on the actual needs in cooperative engineering projects [39]. In the process of engineering project cooperation, due to the long construction period and many participating subjects, the green technology innovation cooperation is more uncertain and unstable, and an external spillover phenomenon is more likely to occur. It is generated by voluntary sharing of knowledge, technology, and other resources, which can promote mutual learning between each other and improve the efficiency of green technology innovation. However, knowledge, technology, and other innovation resources may also be learned and imitated by negative cooperative enterprises or other enterprises through involuntary diffusion, leading to the damage of active cooperative enterprises' own interests. Besides, the enterprises' motivation for green technology innovation can also be easily hit by the opportunistic behavior generated by spillover.

Due to the influence of the above-mentioned construction industry characteristics, construction enterprises do not have complete rationality in the process of green technology innovation cooperation. Therefore, they cannot make the most correct strategy choice at the early stage. They have to make strategy adjustments in the later stage of repeated games until finding the optimal behavior strategy. Based on the above analysis on characteristics of the green technology innovation cooperation among construction enterprise., the following hypotheses are made.

**Hypothesis 1 (H1).** *The influencing factors in the process of cooperation between construction enterprises will lead to the formation of two behavioral strategy choices: positive cooperation and negative cooperation.*

**Hypothesis 2 (H2).** *The percentage of people in enterprise 1 who are willing to actively engage in green technology innovation cooperation is $x$, and the percentage of people of the opposite type is $1 - x$; the percentage of people in enterprise 2 who are willing to actively engage in green technology innovation cooperation is $y$, and the percentage of people of the opposite type is $1 - y$.*

**Hypothesis 3 (H3).** *Both partners have unique knowledge and technological capabilities that can be learned by each other in green technology innovation cooperation. Spillover effects that are formed by the sharing of information and resources in the cooperation process have both positive and negative effects and can affect the benefits of green technology innovation cooperation.*

**Hypothesis 4 (H4).** *A good cooperation reputation can positively affect the benefits of green technology innovation cooperation to a certain extent. On the contrary, enterprises with a poor cooperation reputation will be punished as a result.*

### 2.2. Revenue Matrix

Based on the above model assumptions, construction enterprises will all receive a net income R when they individually engage in green technology innovation. When construction enterprises adopt the cooperative model, the benefits of cooperation are distributed in proportion to the contribution of production factors [40]. The proportion of benefit distribution is assumed to be $\gamma$ for one party and $1 - \gamma$ for the other party. Sinanerzurumlu (2010) showed that sharing in the cooperation process would bring positive spillover effects [41]. Assuming that the positive spillover effect is $\alpha(\alpha > 1)$, such as improving the efficiency of green technology innovation, a higher value of $\alpha$ indicates the higher benefit from cooperation. The study by Oerlemans et al. (2001) showed that sharing in the cooperation process would bring negative spillover effects [42]. Assuming that the negative spillover effect is $\beta(0 < \beta < 1)$, such as causing the leakage of the enterprise's core technology, the higher value of $\beta$ indicates the greater damage to the enterprise's interests. When both parties adopt a positive cooperation strategy, the net benefit is $\alpha\beta\gamma R$ or $\alpha\beta(1 - \gamma)R$. When one party cooperates actively and the other party cooperates negatively, the generation of spillover in the cooperation process makes the negative cooperating party able to gain by learning the knowledge and technology of the positive cooperating party, and at the same time, it can avoid its loss. At this time, there only exists a positive spillover effect, and the net income $\alpha\gamma R$ is finally obtained. The positive partner cannot obtain the positive effect brought by the other party's spillover due to the other party's negative cooperation and has to bear the loss of knowledge and technology assets caused by its spillover; thus, it can only obtain the net income $\beta(1 - \gamma)R$; if both parties adopt negative cooperation, no spillover effect will be generated, and thus the net income R will be obtained. For the party with negative cooperation, although it obtains part of the knowledge and technology of the partner, it needs to pay the liquidated damages E because it fails to fulfill the responsibility of cooperation, and in addition, based on the reciprocity theory [43], such dishonest behavior will affect the reputation of enterprise cooperation [44] and may receive indirect punishment $a$, such as losing preferential tax policies and subsidies on the part of the government [45]. In addition, on the enterprise side, mistrust leads to higher requirements and costs for subsequent cooperation. On the contrary, for enterprises that actively cooperate and comply with the terms of cooperation, their honest behavior can gain the goodwill and trust of the government and enterprises, thus reducing the difficulty of obtaining policy or financial support, and thus receiving indirect income $a$ expected by the government and other enterprises.

Based on the above analysis, a benefit matrix was created, as shown in Table 1. Table 2 shows the definition of parameters.

**Table 1.** Green technology innovation cooperation game income matrix for construction enterprises.

|  | Positive Cooperation | Negative Cooperation |
|---|---|---|
| Positive Cooperation | $\alpha\beta\gamma R + a$, $\alpha\beta(1 - \gamma)R + a$ | $\beta\gamma R + a + E$, $\alpha(1 - \gamma)R - a - E$ |
| Negative cooperation | $\alpha\gamma R - a - E$, $\beta(1 - \gamma)R + a + E$ | $R - a$, $R - a$ |

**Table 2.** Parameter definitions.

| Parameter | Definition |
|:---:|:---:|
| $R$ | net income |
| $\gamma$ | income distribution |
| $\alpha$ | the positive spillover coefficient |
| $\beta$ | the negative spillover coefficient |
| $E$ | the liquidated damages |
| $a$ | indirect income or punishment |
| $x$ | the percentages of positive cooperation in construction enterprise 1 |
| $y$ | the percentages of positive cooperation in construction enterprise 2 |
| $u_1$ | the positive cooperation income of construction enterprise 1 |
| $u_{11}$ | the negative cooperation income of construction enterprise 1 |
| $u_{12}$ | the expected income of construction enterprise 1 |
| $u_2$ | the positive cooperation income of construction enterprise 2 |
| $u_{21}$ | the negative cooperation income of construction enterprise 2 |
| $u_{22}$ | the expected income of construction enterprise 2 |
| $F(x)$ | replication dynamics equation for construction enterprise 1 |
| $G(y)$ | replication dynamics equation for construction enterprise 2 |

*2.3. Replicator Dynamics Equation*

In the process of green technology innovation cooperation in construction enterprises, the percentages of positive cooperation in construction enterprises 1 and 2 are $x$ and $y$, and the percentages of the opposite types of people are $1 - x$ and $1 - y$.

In the process of green technology innovation cooperation in construction enterprises, the positive cooperation income, negative cooperation income and expected income in construction enterprise 1 are $u_{11}$, $u_{12}$, and $u_1$, respectively.

$$u_{11} = y(\alpha\beta\gamma R + a) + (1 - y)(\beta\gamma R + a + E) \tag{1}$$

$$u_{12} = y(\alpha\gamma R - a - E) + (1 - y)(R - a) \tag{2}$$

$$u_1 = xu_{11} + (1 - x)u_{12} \tag{3}$$

In the process of green technology innovation cooperation in construction enterprises, the positive cooperation income, negative cooperation income and expected income in construction enterprise 2 are $u_{21}$, $u_{22}$, and $u_2$, respectively.

$$u_{21} = x[\alpha\beta(1 - \gamma)R + a] + (1 - x)[\beta 1 - \gamma R + a + E] \tag{4}$$

$$u_{22} = x[\alpha(1 - \gamma)R - a - E] + (1 - x)(R - a) \tag{5}$$

$$u_2 = yu_{21} + (1 - y)u_{22} \tag{6}$$

The replicated dynamic equations for construction enterprises 1 and 2 are calculated as:

$$F(x) = \frac{dx}{dt} = x(u_{11} - u_1) = x(1 - x)[y(\alpha\beta\gamma R - \beta\gamma R - \alpha\gamma R + R) + \beta\gamma R - R + 2a + E] \tag{7}$$

$$G(y) = \frac{dy}{dt} = y(u_{21} - u_2) = y(1 - y)\{x[(1 - \gamma)(\alpha\beta R - \beta R - \alpha R) + R] + \beta(1 - \gamma)R - R + 2a + E\} \tag{8}$$

### 3. Model Analysis

#### 3.1. Replicator Dynamics Equation Analysis

Let replicator dynamics equation $F(x) = 0$ and $G(y) = 0$, and calculate the five equilibria points of this game model on the two-dimensional plane $\{( x, y), 0 \leq x \leq 1, 0 \leq y \leq 1\}$, denoted as $E_1(0,0)$, $E_2(1,0)$, $E_3(0,1)$, $E_4(1,1)$, $O(x_0, y_0)$, as shown in Figure 1. In Figure 1, I, II, III, IV is the region divided by $E_1 (0,0)$, $E_2 (1,0)$, $E_3 (0,1)$, $E_4 (1,1)$, $O(x_0, y_0)$.

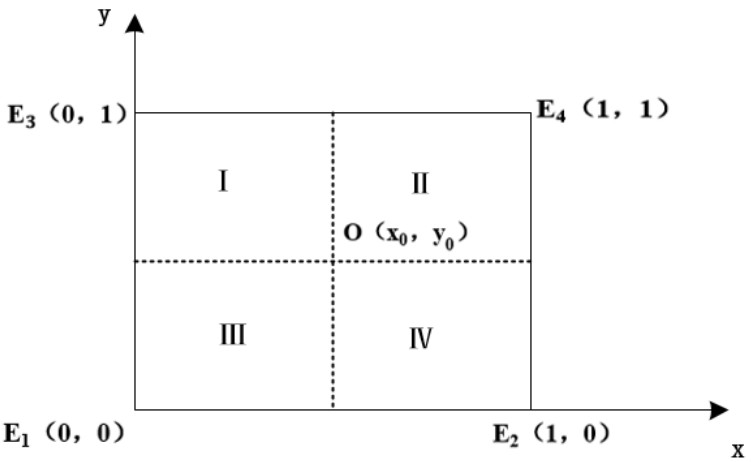

**Figure 1.** Diagram of the evolutionary game interval and equilibrium point.

The equilibrium point $O(x_0, y_0)$: this equilibrium point is a saddle point, where $x_0 = -\frac{\beta(1-\gamma)R - R + 2a + E}{(1-\gamma)(\alpha\beta R - \beta R - \alpha R) + R}$ and $y_0 = -\frac{\beta\gamma R - R + 2a + E}{\alpha\beta\gamma R - \beta\gamma R - \alpha\gamma R + R}$. When construction enterprises 1 and 2 are at this equilibrium point, the choice of their cooperation strategy is uncertain, and the final evolutionarily stable strategy can be formed after continuous game play.

#### 3.2. Analysis on the Evolutionarily Stable Strategies of Each Construction Enterprise

First, the evolutionarily stable strategy of construction enterprise 1 is analyzed, and after the first-order derivation of the replicator dynamics equation of construction enterprise 1, the following equation is further organized.

$$F(x)' = (1 - 2x)[y(\alpha\beta\gamma R - \beta\gamma R - \alpha\gamma R + R) + \beta\gamma R - R + 2a + E] \tag{9}$$

When $-\beta\gamma R + R - 2a - E > \alpha\beta\gamma R - \beta\gamma R - \alpha\gamma R + R$, $F(x)' < 0$ holds constantly; thus, $x_1 = 0$ is the evolutionarily stable strategy of construction enterprise 1; when $x = 1$, $F(x)' > 0$ holds constantly; thus, $x_2 = 1$ is not the evolutionarily stable strategy for construction enterprise 1.

Based on the above results, the case (1) can be obtained: when $-\beta\gamma R + R - 2a - E > \alpha\beta\gamma R - \beta\gamma R - \alpha\gamma R + R$, and $y$ is in any range of values, $x_1 = 0$ is the evolutionarily stable strategy for construction enterprise 1.

Similarly, analyzing when $-\beta\gamma R + R - 2a - E < \alpha\beta\gamma R - \beta\gamma R - \alpha\gamma R + R$, and come to case (2): when $y > y_0$, since $F'(x_2 = 1) < 0$, $x_2 = 1$ is the evolutionarily stable strategy of construction enterprise 1; when $y < y_0$, due to $F'(x_1 = 0) < 0$, $x_1 = 0$ is the evolutionarily stable strategy of construction enterprise 1.

Similarly, analyzing the construction enterprise 2, and come to the case (3): when $-\beta(1 - \gamma)R + R - 2a - E > (1 - \gamma)(\alpha\beta R - \beta R - \alpha R) + R$, $y_1 = 0$ is the evolutionarily stable strategy of construction enterprise 2.

Then, case (4) can be obtained: when $-\beta(1 - \gamma)R + R - 2a - E < (1 - \gamma)(\alpha\beta R - \beta R - \alpha R) + R$, and $x > x_0$, since $G'(y_2 = 1) < 0$, $y_2 = 1$ is the evolutionarily stable strategy of construction enterprise 2; when $x < x_0$, since $G(y_1 = 0) < 0$, therefore $y_1 = 0$ is the evolutionarily stable strategy for construction enterprise 2.

### 3.3. Analysis on the Evolutionarily Stable Strategy between the Two Sides of Construction Enterprises

In the actual cooperation process, the two sides of the enterprise will simultaneously carry out a dynamic game. The results of the evolutionarily stable strategy may change due to the interaction of the two sides of the game. The combination of strategic selection in different situations will produce four results.

The first result: when construction enterprise 1 and construction enterprise 2 are in case (1) and case (3), respectively, to select cooperation strategies, the evolutionary game results of both sides are shown in Figure 2. In this case, negative cooperation of construction enterprises 1 and 2 is the final evolutionarily stable strategy.

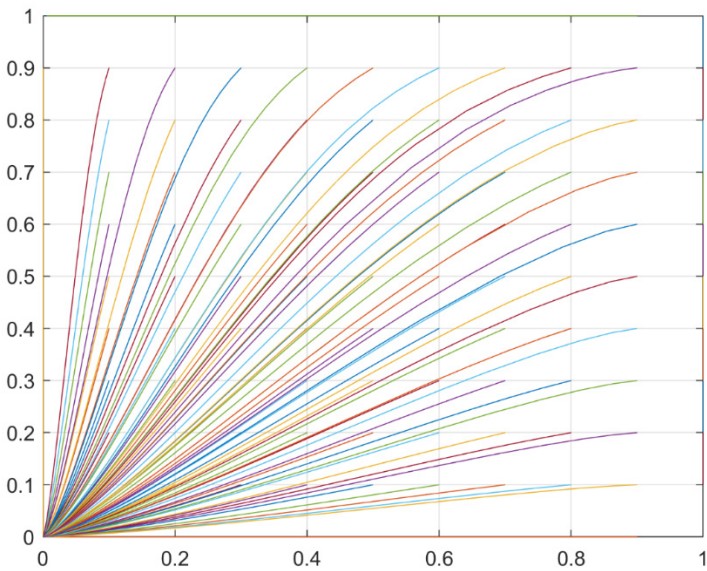

**Figure 2.** Dynamic evolution phase diagram for construction enterprises 1 and 2 in case (1) and (3).

The second result: when construction enterprise 1 and construction enterprise 2 are in case (1) and case (4), respectively, to select cooperation strategies, the evolutionary game results are shown in Figure 3. In this case, negative cooperation of construction enterprises 1 and 2 is the final evolutionarily stable strategy.

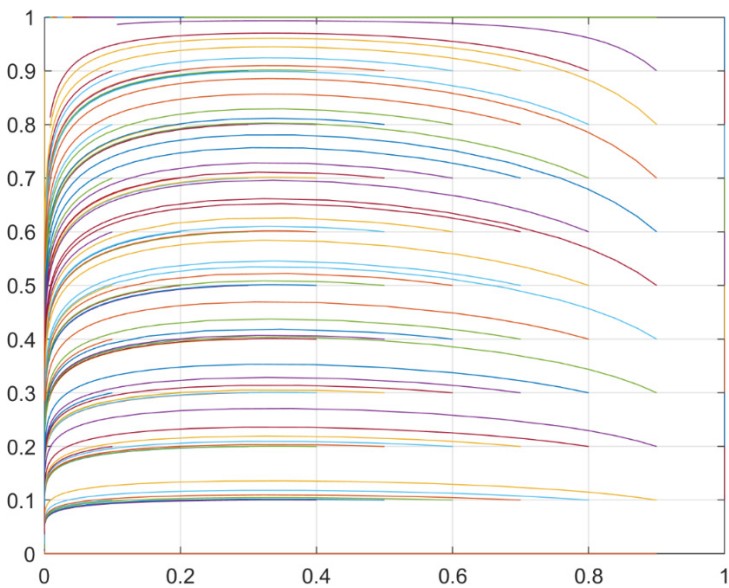

**Figure 3.** Dynamic evolution phase diagram for construction enterprises 1 and 2 in case (1) and (4).

The third result: when construction enterprise 1 and construction enterprise 2 are in case (2) and case (3), respectively, to select cooperation strategies, the evolutionary game results are shown in Figure 4. In this case, negative cooperation of construction enterprises 1 and 2 is the final evolutionarily stable strategy.

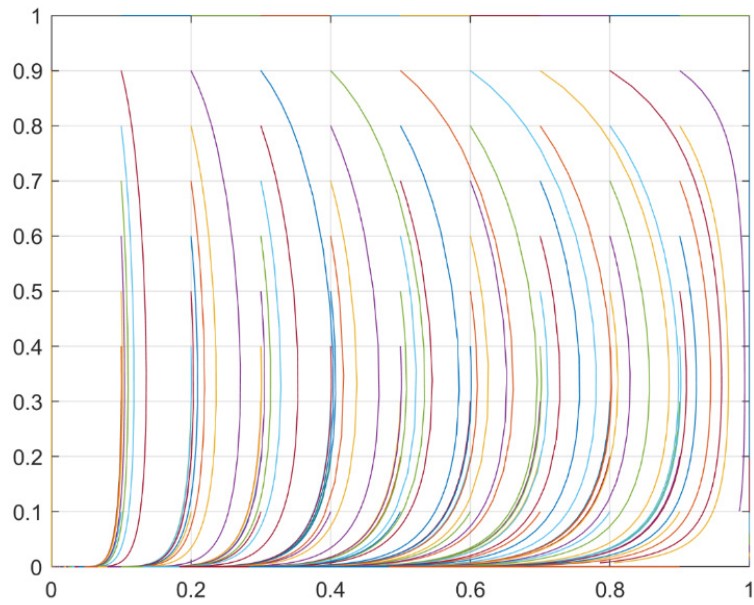

**Figure 4.** Dynamic evolution phase diagram for construction enterprises 1 and 2 in case (2) and (3).

The fourth result: when construction enterprise 1 and construction enterprise 2 are in case (2) and case (4), respectively, to select cooperation strategies, the evolutionary game results of both sides are shown in Figure 5. In this case, evolutionarily stable strategy for construction enterprises 1 and 2 has uncertainty, which may both choose negative cooperation or positive cooperation, and is related to the location of the saddle point, where the probability of evolving to the negative cooperation strategy is $\frac{(x_0+y_0)}{2}$.

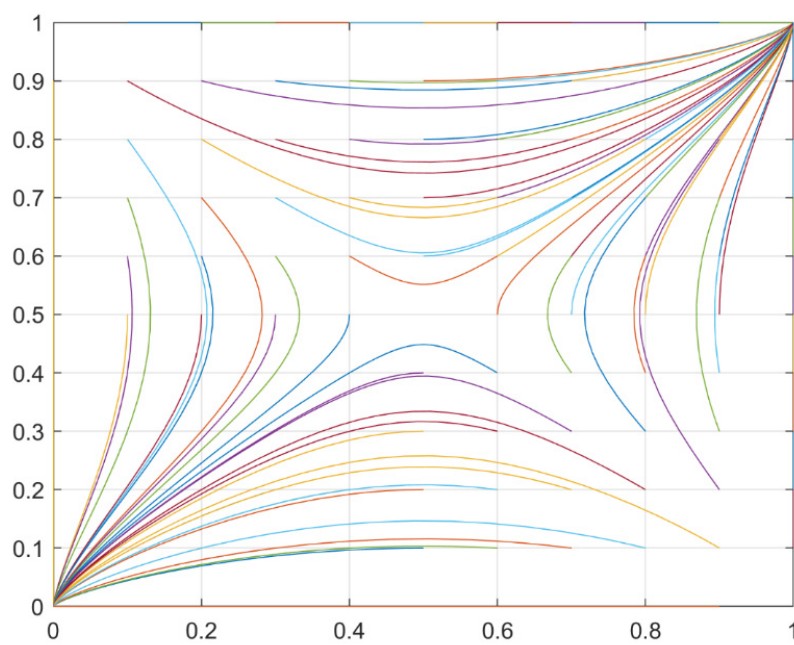

**Figure 5.** Dynamic evolution phase diagram for construction enterprises 1 and 2 in case (2) and (4).

The above results are summarized and analyzed to conclude that when the inequalities $-\beta(1-\gamma)R + R - 2a - E < (1-\gamma)(\alpha\beta R - \beta R - \alpha R) + R$ and $-\beta\gamma R + R - 2a - E < \alpha\beta\gamma R - \beta\gamma R - \alpha\gamma R + R$ both hold, {positive cooperation, positive cooperation} may become evolutionarily stable strategy for construction enterprises 1 and 2. The value of the saddle point $O(x_0, y_0)$ affects the final evolutionarily stable strategy choice of the cooperation parties. If the area occupied by the positive cooperation region $E_4$-$E_2$-$E_3$-O is large, the partners will tend to select the positive cooperation strategy. On the contrary, when the area occupied by the positive region $E_1$-$E_2$-$E_3$-O becomes smaller, the partners will tend to select the negative cooperation strategy just as the value of the saddle point becomes larger.

## 4. Numerical Simulation

From the conclusion of Section 3.3, only if the inequalities $-\beta(1-\gamma)R + R - 2a - E < (1-\gamma)(\alpha\beta R - \beta R - \alpha R) + R$ and $-\beta\gamma R + R - 2a - E < \alpha\beta\gamma R - \beta\gamma R - \alpha\gamma R + R$ both hold, {positive cooperation, positive cooperation} may become evolutionarily stable strategy for construction enterprises 1 and 2. This section is to further explore how the influencing factors promote green technology innovation of construction enterprises to adopt positive cooperation strategy. Based on the evolutionary game model established in the previous section, numerical simulation is carried out by matlab2018b software. The number of simulations is set to 20, and the initial assignment of parameters corresponding to each influencing factor is shown in Table 3, which can satisfy the two inequalities.

**Table 3.** Model parameter values.

| Parameters | $R$ | $\gamma$ | $\alpha$ | $\beta$ | $E$ | $a$ |
|---|---|---|---|---|---|---|
| values | 10 | 0.45 | 1.5 | 0.6 | 2 | 1.5 |

In order to objectively analyze the role of each influencing factor on the choice of cooperative behavior strategies when studying the law of the role of influencing factors other than cooperation willingness, the initial proportion of the positive cooperation population is all set at 0.5.

### 4.1. Cooperation Willingness

In this paper, the proportion $y$ of the cooperation willing in construction enterprise 2 is chosen to analyze the effect on the proportion $x$ of cooperation willing in construction enterprise 1. Two values in the range of about 0.5 are selected for the analysis, such that the proportion $y$ of cooperation willing in construction enterprise 2 are 0.3 and 0.7, respectively. As shown in Figure 6a, when $y = 0.3$, the value of $x$ finally converges to 0. As shown in Figure 6b, when $y = 0.7$, the value of $x$ finally converges to 1. This reflects that the strong cooperation willingness for construction enterprise 2 will have a positive impact on the cooperation willingness for construction enterprise 1 and make it adopt a positive cooperation strategy.

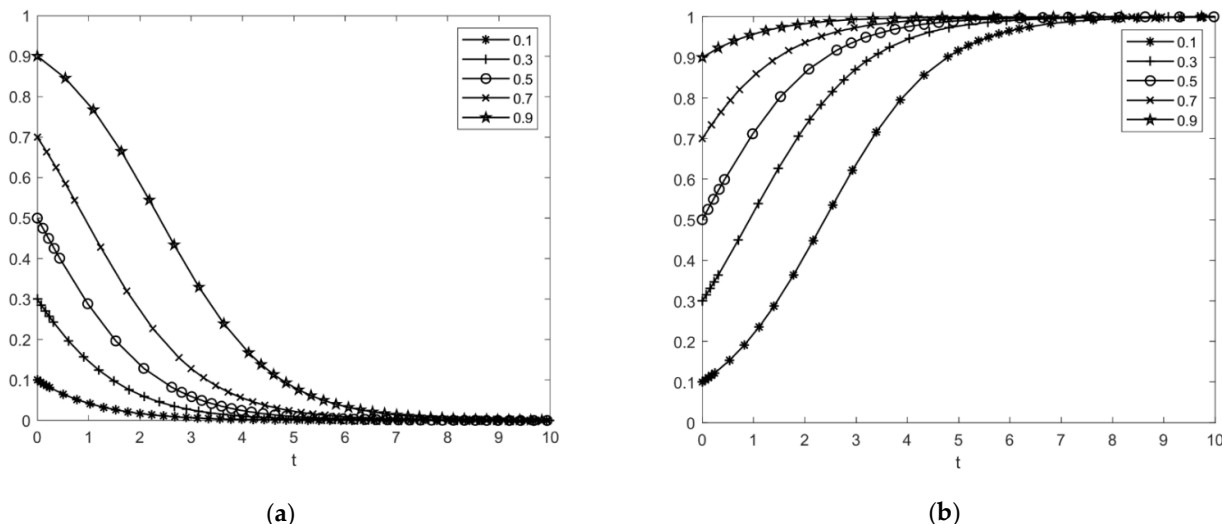

**Figure 6.** Cooperation willingness. (**a**) $y = 0.3$, (**b**) $y = 0.7$.

### 4.2. Sharing Level

The spillover effect has both positive and negative sides. The relative magnitudes of the spillover coefficients $\alpha$ and $\beta$ are selected to indicate the effects on the evolutionarily stable strategies of cooperation parties. Analyzing construction enterprise 1, as shown in Figure 7, when the negative spillover coefficient $\beta$ is 0.3 and the positive enterprise spillover coefficient $\alpha$ is greater than a critical value between 1.7 and 1.9, the higher the value of $\alpha$, the greater the possibility of positive cooperation for construction enterprises. Conversely, the lower the value of $\alpha$, the greater the possibility for negative cooperation for construction enterprises. When the negative spillover $\beta$ is 0.7 and the positive spillover $\alpha$ is greater than a critical value between 1.9 and 2.1, the higher the value of $\alpha$, the higher the possibility of positive cooperation for construction enterprises. Conversely, the lower the value of $\alpha$, the greater the possibility of negative cooperation for construction enterprises. The above analysis shows that when the additional income from positive spillover is greater than the loss from negative spillover, the construction enterprises with evolutionarily stable strategies will tend to select the positive cooperation strategy, and conversely, more likely to select the negative cooperation strategy.

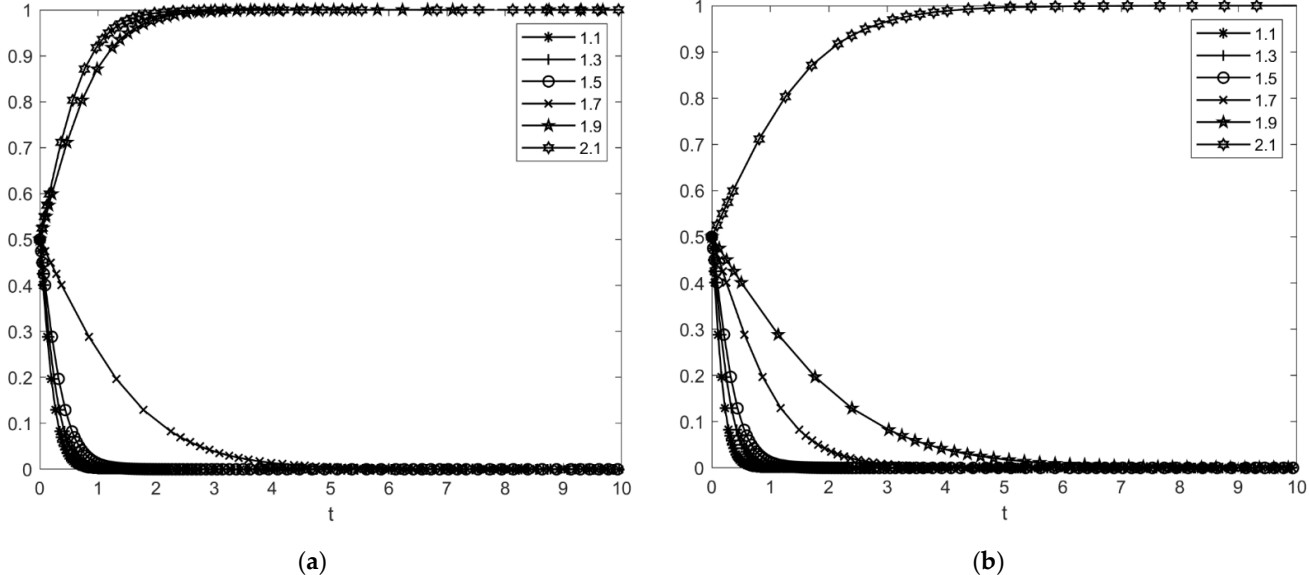

**Figure 7.** Sharing Level. (**a**) $\beta = 0.3$ (construction enterprise 1), (**b**) $\beta = 0.7$ (construction enterprise 1).

### 4.3. Income Distribution

The evolutionarily stable strategies of construction enterprises 1 and 2 with different income distributions are shown in Figure 8. When the income distribution coefficient $\gamma$ is greater than a critical value between 0.3 and 0.5, the larger the value, the more possibly construction enterprise 1 tends to cooperate positively and construction enterprise 2 tends to cooperate negatively. When the income distribution coefficient $\gamma$ is less than the critical value, the smaller the value, the more possibly construction enterprise 2 tends to cooperate positively and construction enterprise 1 tends to cooperate negatively. In summary, the more income they obtain from cooperation, the more possibly the cooperation stable strategies for construction enterprises evolve in the direction of positive cooperation. However, the other party will think that the income they obtain is too small, producing a sense of unfairness, and they will think that the income obtained from cooperation is not equal to the initial investment, which leads to the reduction of their cooperation enthusiasm.

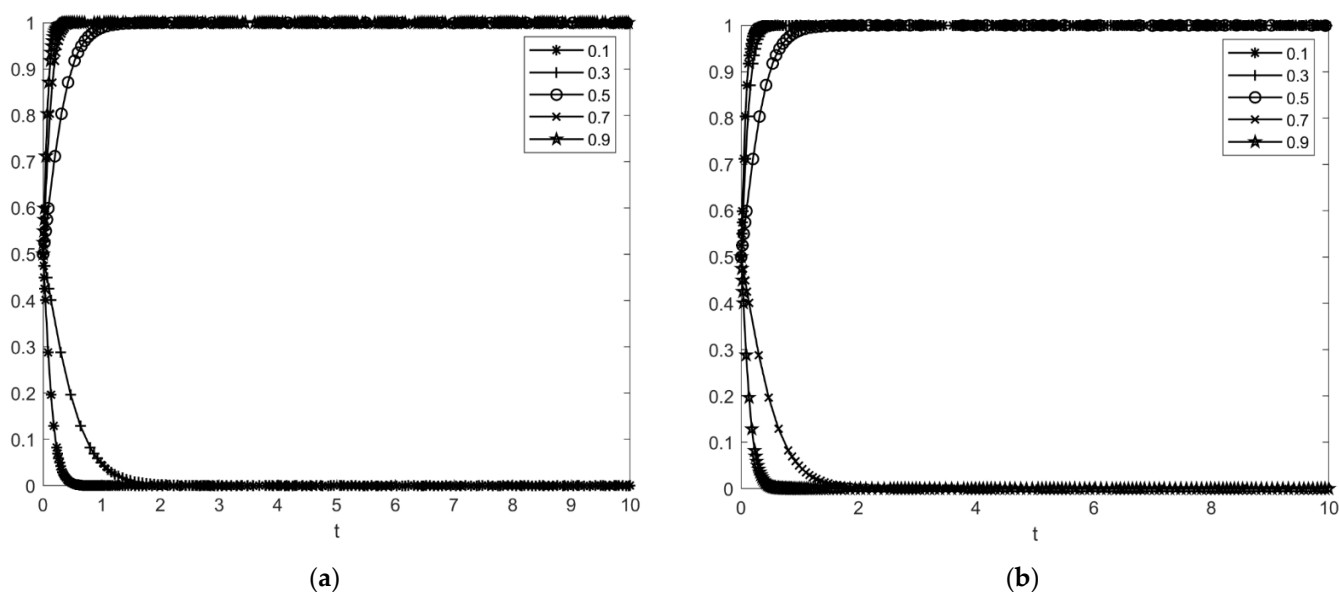

(**a**)																																							(**b**)

**Figure 8.** The effect of $\gamma$ on the cooperation strategy for construction enterprises 1 and 2. (**a**) construction enterprise 1; (**b**) construction enterprise 2.

### 4.4. Punishment Mechanism

The $2a + E$ that appears in the replication dynamic equation is treated as a parameter indicating the effect of the punishment mechanism. The effect of the value of it on the choice of cooperation strategy is analyzed. The evolutionarily stable strategies of construction enterprises 1 and 2 are shown in Figure 9. When $2a + E$ is greater than a certain critical value between 4.5 and 6.0, the larger its value, the more the evolutionarily stable strategies of construction enterprises 1 and 2 tend to cooperate positively; when $2a + E$ is less than this critical value, the smaller its value, the more the evolutionarily stable strategies of construction enterprises 1 and 2 tend to cooperate negatively. In summary, because opportunistic behavior requires a large cost of punishment, the more enterprises tend to select the evolutionarily stable strategy of positive cooperation when the punishment mechanism is more perfect and the punishment is more effective within a reasonable range. They think their own interests can be guaranteed under this condition.

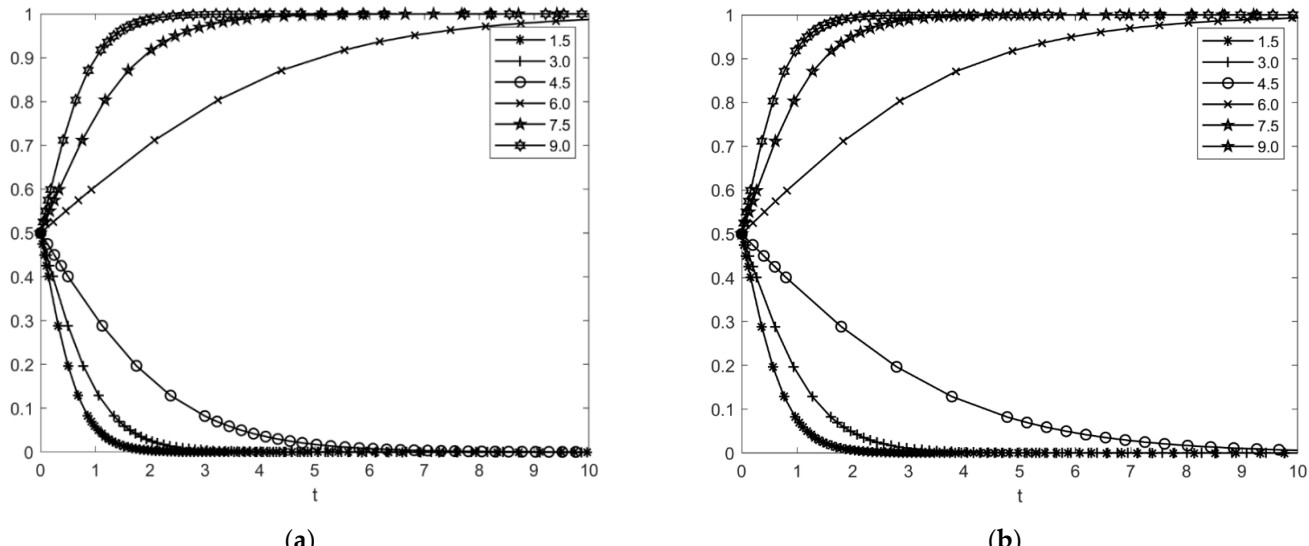

**Figure 9.** The effect of $2a + E$ on the cooperation strategy of enterprises 1 and 2. (**a**) enterprise 1; (**b**) enterprise 2.

## 5. Conclusions, Limitations and Future Research

### 5.1. Conclusions

Based on the results of numerical simulation, the willingness to cooperate, sharing level, income distribution and punishment mechanism have important effects on the green technology innovation cooperation of construction enterprises. Specifically, first, the willingness of one side of construction enterprises to cooperate affects the cooperation strategy choice of the other side, and the higher the willingness to cooperate, the more the cooperation sides tend to positively cooperate, which is conducive to promoting the success of green technology innovation cooperation. Second, the higher the level of information, knowledge, technology and other resource sharing, the more the cooperation parties tend to positively cooperate under the effect of a positive spillover effect, and the more likely the enterprise green technology innovation will achieve results. On the contrary, if there is a lack of effective supervision and intellectual property protection in the process of adequate communication and exchange between enterprises, the negative spillover effect will easily lead to opportunistic behavior, and the cooperation results between the two parties will hardly lead to success. Third, the higher the coefficient of own income distribution, the higher the possibility of enterprises choosing positive cooperation strategies. Last, within a reasonable range, the stronger the punishment, the better the punishment effect, the more it can combat the opportunistic behavior of enterprises in the cooperation process, thus increasing the enthusiasm of enterprises to choose the green technology innovation cooperation strategy, and the greater the possibility of both partners to choose the positive cooperation strategy together.

In summary, in order to promote the improvement of green technology innovation in the construction industry and avoid opportunistic behavior in the cooperation process, the paper proposes the following suggestions for the government and the cooperation construction enterprises.

1.  Improve the cooperation willingness and emphasis of enterprises. The government should introduce and improve the relevant supporting policies, regulations and systems for green technology innovation cooperation. The construction enterprises should establish target management and incentive mechanisms in the cooperation process.
2.  Strengthen communication among enterprises and improve the sharing level. Construction enterprises should establish a communication system, set up a good channel

for information exchange to convey solutions to problems and information in a timely manner such as regular talks and research.

3. Enhance intellectual property protection to protect the interests of construction enterprises. In the process of adequate communication and information exchange, the enterprises should have a basic awareness of intellectual property protection for the confidential and important technologies, commercial brands and other intangible assets. Meanwhile, the government should establish an effective monitoring and intellectual property protection system.

4. Optimize the benefit distribution mechanism and enhance the sense of fairness of enterprises. Cooperation enterprises should strengthen communication and consultation on the scope, proportion and manner of income distribution before establishing a cooperative relationship, such that a mutually satisfactory and reasonable income distribution plan can be determined in cooperation.

5. Establish an assessment and evaluation mechanism to implement penalties. Construction enterprises should regularly assess the performance and evaluate the contribution degree of cooperation partners. In addition, the construction industry can establish a cooperation credit system to regularly assess the cooperation performance of construction enterprises. From there, penalties such as charging default fees or raising the cooperation deposit can be imposed on construction enterprises that adopt opportunistic behavior.

### 5.2. Limitations and Future Research

The findings of this study are valuable, but there are limitations and shortcomings that require more in-depth research. First, this study was conducted on construction enterprises in China, and the scope of the study could be extended to other countries. Second, as this paper has not conducted any research on the differences of different cooperation modes in the research of green technology innovation cooperation of construction enterprises, the research can be further refined in the future.

**Author Contributions:** Conceptualization, W.L. and Q.W.; methodology, W.L. and Q.W.; software, W.L.; validation, Q.W. and W.L.; formal analysis, Q.W. and W.L.; resources, Q.W.; data curation, M.D.; writing—original draft preparation, W.L. and M.D.; writing—review and editing, Q.W. and Q.Q.; funding acquisition, Q.W.; reviewing and editing, Q.W. and Q.Q. All authors have read and agreed to the published version of the manuscript.

**Funding:** This research was funded by the National Natural Science Foundation of China (grant number 71942006 and 72171237).

**Institutional Review Board Statement:** Not applicable.

**Informed Consent Statement:** Not applicable.

**Data Availability Statement:** Exclude this statement.

**Acknowledgments:** The authors are grateful for comments and recommendations from the editor and anonymous reviewers.

**Conflicts of Interest:** The authors declare no conflict of interest.

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
