# Peer review of "Research on Cooperative Behavior of Green Technology Innovation in Construction Enterprises Based on Evolutionary Game"

_buildings, doi:10.3390/buildings12010019_

Round 1

Reviewer 1 Report

The Authors build a dynamic evolution game model and analyzes the evolutionary stable strategy by numerical simulation of various kinds of influence factors for building enterprise effect of the green technology innovation cooperation strategy selection. The topic is really interesting, but some parts of the paper have to be improved.

Following, I report the main points:

  • Row 84: It is important to specify and introduce a deep literature review in order to claim that “there is a lack of cooperation research related to green technology innovation in the construction field”.
  • Why in this paper you talk about the enterprise green technology innovation?
  • How do you choose the parameters in table 2?

I report the minor points:

  • Abstract: re-write the abstract (some capital letter in the text, …)
  • Row 92: with enterprise- enterprise
  • Row 131: Translated with www.DeepL.com/Translator (free version)Having reciprocity pref- 131 erences.

By considering the different points, I suggest major revision of the paper.

Reviewer 2 Report

The authors have conducted a research on the cooperative behavior of green technology innovation in construction enterprises based on the evolutionary game. The topic is interesting; however, the manuscript needs major revision to be considered for publication in journal of buildings. My comments are as follows.

  • In the abstract, I cannot see where the sentences are finished. There are some words with capital letters in the text. The authors have to proofread the whole abstract carefully.
  • Overall, try to write short sentences to make it easy-to-read. For instance, the first sentence in the introduction can be split into two sentences. Please check the whole manuscript.
  • Also, the paragraphs are too long. For instance lines 47-80. The authors may consider shortening the length of the paragraphs or splitting them into 2-3 paragraphs.
  • Line 31, what is “so on”?
  • Line 82-83, how did you summarize the results? What do you mean by home? China? Please revise.
  • Line 81-108, present the research gaps that the authors have identified using numbers or bullet points.
  • Line 110, avoid using I, we, etc.
  • Line 131, what is this? “Translated with www.DeepL.com/Translator (free version)”
  • Line 175, what analysis?
  • Make sure all symbols and characters are defined in the text. For instance, what is ?′  ? if necessary, provide a nomenclature.
  •  
  • The last section must be the conclusion of the research which is missing in this manuscript.
  • The authors must review the current literature on this topic. I cannot see any reference published in 2021.

Round 2

Reviewer 1 Report

no other notes

Reviewer 2 Report

The authors have addressed my comments and I recommend this manuscript for publication